# Energy System Contributions during Olympic Combat Sports: A Narrative Review

**DOI:** 10.3390/metabo13020297

**Published:** 2023-02-17

**Authors:** Emerson Franchini

**Affiliations:** Martial Arts and Combat Sports Research Group, Sport Department, School of Physical Education and Sport, University of São Paulo, São Paulo 05508-030, Brazil; efranchini@usp.br; Tel.: +55-11-3091-2124

**Keywords:** glycolysis, oxidative system, phosphagens system, phosphocreatine, martial arts, performance

## Abstract

This narrative review focuses on the studies that estimate the energy systems’ contributions during match simulations of striking (boxing, karate, and taekwondo), grappling (judo), and weapon-based (fencing) Olympic combat sports. The purpose is to provide insights into the metabolism of these athletes. In striking Olympic combat sports, the oxidative contribution varied from 62% (in karate and taekwondo) to 86% (in boxing), the ATP-PCr system contribution varied from 10% (in boxing) to 31% (in taekwondo), and the glycolytic contribution was between 3% (in the third round of taekwondo) and 21% (in karate). In grappling combat sports, only judo was studied, and for a 4 min match, the oxidative contribution was 79%, followed by 14% ATP-PCr system contribution and 7% contribution from the glycolytic system. In fencing, the only weapon-based Olympic combat sport, the oxidative contribution varied from 81% (in the first bout) to 90% (in the second bout), followed by 9% (bout 2) to 12% (bout 1) contribution from the ATP-PCr system, and 0.6% to 7% contribution from the glycolytic system during 3 × 3 min bouts of épée match simulation. Hence, Olympic combat sports are primarily powered by the oxidative system, but the key scoring actions are likely fueled by anaerobic pathways.

## 1. Introduction

Combat sports represented 26% of all medals in the 2020 Tokyo Olympics [1]. Professional combat sports (e.g., boxing and MMA) are growing in popularity and attracting more spectators [2]. These sports include striking (e.g., boxing, karate, taekwondo), grappling (e.g., judo, Greco-Roman and freestyle wrestling), mixed (e.g., hapkido, MMA), and weapon-based (e.g., fencing, kendo) sports, and their performance is influenced by multiple physiological factors [3]. Thus, understanding the physiological responses of each combat sport is important for improving training and athlete performance [4].

Some combat sports such as judo, karate, kung fu, and taekwondo have both forms (e.g., kata in karate and judo, taolu in kung fu, poomse in taekwondo) and combat, with rules regulating the type and intensity of contact [5,6,7]. At the Olympics, only karate has forms events (kata) [8], whereas the other combat sports only have combat events [1]. Measuring physiology during forms is easier since equipment does not significantly limit athlete movements. However, using portable equipment during combat matches is prohibited as it may cause injury or damage to equipment due to the high level of contact [9]. Researchers have overcome this issue by simulating matches [10,11,12,13,14,15,16,17,18], developing sport-specific tests to simulate match’s physiological responses [19,20,21,22,23,24], determining effort-pause ratios to infer the physiological and metabolic demands [11,25,26,27,28,29,30,31,32], or assessing physical capacities of athletes with different competitive levels [33,34,35,36].

Studies have reviewed combat-sport-specific tests [37], time-motion analysis [26,38,39,40], physiological characteristics of athletes [5,6,7,39,40,41,42,43], and performance responses to repeated matches [44]. However, no review was found on the energy systems’ contributions to combat sports matches, simulations, or competitions. This information is crucial for coaches, sport scientists, and strength and conditioning professionals to improve training programs for high-level athletes [45]. Briefly, three energy systems contribute to adenosine triphosphate (ATP) resynthesis and thus to energy release during exercise, as only the hydrolysis of the ATP molecule provides energy for skeletal muscle contraction. These systems are the ATP-PCr (adenosine triphosphate and phosphocreatine), glycolytic, and oxidative systems [46]. The ATP-PCr and glycolytic systems are classified as anaerobic, whereas the oxidative system is classified as aerobic. The ATP-PCr system has a higher metabolic power (i.e., rate of energy transfer per unit of time) due to a low number of reactions needed to resynthesize ATP, but a low capacity (i.e., total amount of energy that can be released) due to limited substrate stores. The glycolytic system resynthesizes ATP through the non-aerobic breakdown of carbohydrates, mainly stored as muscle glycogen, and has intermediate metabolic power due to the higher number of reactions involved compared to the ATP-PCr system, but a greater capacity due to a large amount of stored carbohydrates. However, it is mainly limited by the accumulation of metabolites during activation. The oxidative system depends on oxygen consumption and degrades carbohydrates, fats, and sometimes protein. It has the lowest metabolic power among the energy systems due to many reactions involved in the oxidative breakdown of these substrates, but it has the highest capacity as the stores of these substrates are abundant. The activation of these energy systems determines the rate of energy release and thus the intensity and duration of effort [46]. This narrative review focuses on presenting the energy systems contributions during Olympic combat sports (boxing, fencing, judo, karate, taekwondo, and wrestling) match simulations and does not consider studies using isolated tests or actions outside the combat context. Additionally, as no study was found that investigated the energy system contributions during simulated wrestling matches using physiological measurements, this review only included studies on boxing, fencing, karate, and taekwondo.

## 2. Energy Systems’ Contributions in Combat Sports

Physical performance is a result of both metabolic power and mechanical efficiency. Improving either of these two factors or a combination of both can enhance performance, i.e., increasing athlete’s metabolic power and/or elevating technical skills (i.e., mechanical efficiency) [24]. Energy release is dependent on the intensity and duration of the effort executed, and the energy optimization to accomplish a given task is determined by the ATP-PCr, the glycolytic and the oxidative system contributions. These energy systems vary in terms of metabolic power and capacity, which are higher in individuals specifically trained to perform better in tasks depending predominantly on a given energy system or a combination of energy systems [46]. Therefore, determining the energy system contributions in a given sport is extremely relevant to provide more specific physical stimuli for the athlete preparing to compete on it, and both diet and tailored ergogenic substances could be prescribed to specifically achieve the athletes’ needs and increase performance.

Different methods have been used to estimate energy system contributions in various sports and physical activities. The main methods used are muscle biopsy [47], oxygen deficit method [48], oxygen uptake, fast component of excess oxygen consumption (EPOCfast), and blood lactate [49]. Additionally, mathematical modeling [50] is also used. However, each method has its own limitations and assumptions, which are not the focus of this review. Some methods cannot be used to estimate energy system contributions when efficiency or mechanical power/speed cannot be quantified, but others can be used during combat sport match simulations. This method assumes that the increase in oxygen consumption above resting values represents the oxidative contribution, that each 1 mmol/L of accumulated blood lactate is equivalent to 3 mL/kg/min of oxygen consumption and represents the glycolytic system contribution, and that the fast phase of excess post-exercise oxygen consumption represents the ATP-PCr contribution [49]. Furthermore, each 1 L of oxygen consumption is equivalent to 20.92 kJ [46]. The total energy expenditure is the sum of the three systems, and the relative contribution of each system is calculated by dividing the absolute value of a given system by the total energy expenditure [9]. The method involving the measurement of oxygen uptake, EPOCfast, and blood lactate has been the only method used to estimate energy system contributions in combat sports [12,13,14,15,16,18,51,52,53,54]. Moreover, this method does not require quantifying the mechanical output, and it allows estimating the energy systems by measuring continuous oxygen uptake during simulations of combat sport matches [9]. This is particularly relevant because during intense, intermittent efforts, such as those performed in combat sports, the gas exchange is entirely kinetic, meaning that the dynamics of pulmonary oxygen uptake and pulmonary carbon dioxide production never reach a steady state. This because their kinetics are slow relative to the imposed effort [55]. This method has been shown to be capable of terminating the maximum accumulated oxygen deficit, with the advantage of breaking down the anaerobic contribution into its ATP-PCr component (based on the EPOCfast) and glycolytic components (based on the conversion of blood lactate accumulation to oxygen equivalent) [56]. Its use and validity have been reported in running [57,58,59], and cycling [60,61], including high-intensity interval exercise [62].

### 2.1. Striking Olympic Combat Sports

At the last edition of the Olympic Games, boxing, karate, and taekwondo were the striking combat sports contested [1].

Figure 1 shows the relative energy system contributions during simulated karate matches.

The first study using the EPOCfast and blood lactate method during simulated combat analyzed karate kumite [12]. The authors studied 36 matches, with 2 to 4 matches per athlete and a mean duration of 275 ± 61 s with 17, 15, and 9 min intervals between matches for athletes who competed in four matches. They used video recordings of the matches to distinguish between low-intensity actions (e.g., stepping and displacements) and high-intensity actions (e.g., attacks and counterattacks), as well as pauses. The results (Figure 1) showed that the matches had a 77.8 ± 5.8% contribution from the oxidative metabolism, 16.0 ± 4.6% from the ATP-PCr system, and 6.2 ± 2.4% from the glycolytic system, with a total energy expenditure of 341.0 ± 81.9 kJ. These results indicate that karate kumite is predominantly oxidative, but the scoring actions are likely supported by the anaerobic pathways. Positive, significant correlations (*p* < 0.05) were found between the ATP-PCr metabolic power and the net-action rate during the match (r = 0.46), and between the glycolytic metabolic power and the net-action rate (r = 0.49). A negative, significant correlation was also observed between the glycolytic system metabolic power and the number of matches (r = −0.76), and between the total metabolic power and the number of matches (r = −0.46). This suggests that athletes who were better at activating the ATP-PCr system executed more actions during the match and that those who could maintain glycolytic and total metabolic power throughout the competition were more likely to perform a higher number of attacking actions. Future studies should examine interventions (e.g., ergogenic aids, training programs) to determine if improving the anaerobic pathways’ metabolic power leads to an increased number of attacks in karate matches.

Doria et al. [15] also investigated karate athletes performing kumite and reported a predominance of the oxidative system (74 ± 1%), followed by the ATP-PCr (14 ± 3%) and the glycolytic system (12 ± 2%), with a total energy expenditure of 304.8 ± 25.5 kJ (Figure 1). This study also compared male and female karate athletes and found that male athletes had a higher oxidative contribution during the simulated kumite match. The match metabolic power was slightly higher than the maximal oxygen consumption metabolic power determined during a cycle ergometer test, even though the mean oxygen consumption during the match was only 55% of the maximal oxygen consumption measured during the cycle ergometer test. Therefore, based on these findings, kumite matches are predominantly oxidative, but the athletes’ oxygen consumption values achieved during the match are not close to their maximum capabilities, suggesting low cardiovascular stress between successive attacks.

It is relevant to mention that a case study [53] of a double world champion male karate athlete observed a total energy expenditure (307.5 kJ) similar to previous studies [12,15], but with a much higher glycolytic contribution (31%) and a lower oxidative contribution (61%) than previously reported [12,15]. Given that the glycolytic system may play a crucial role in successive high-intensity actions in karate, and that this system’s metabolic power was positively correlated with the net-action rate [12], it is likely that this athlete was able to maintain a higher attack frequency, which could have contributed to his success.

Figure 2 presents the relative energy system contributions during a taekwondo simulated match and for each round.

Taekwondo was also analyzed to determine the energy system contributions [13]. The authors reported that oxidative systems contributed 66 ± 6%, ATP-PCr contributed 30 ± 6%, and glycolytic systems contributed 4 ± 2% during a simulated taekwondo match composed of three 2 min rounds interspersed by 1 min intervals (Figure 2). The mean total energy expenditure in each round was 181 ± 28 kJ (i.e., a total of ~543 kJ) and the mean metabolic power across the rounds was 1.40 ± 0.22 kW. Although the attack time, number of attacks, sum of attack time, and sum of time without attacks remained constant in the three rounds (*p* > 0.05), the energy system contributions varied. Specifically, absolute oxidative contribution was lower (*p* < 0.05) in round 1 (98 ± 15 kJ) compared to rounds 2 (127 ± 14 kJ) and 3 (134 ± 18 kJ), and absolute glycolytic contribution decreased (*p* < 0.05) from round 1 (11 ± 4 kJ) to round 3 (6 ± 5 kJ). The total energy expenditure in the first round (158 ± 17 kJ) was also lower (*p* < 0.05) than in the second (183 ± 17 kJ) and third (203 ± 29 kJ) rounds. Thus, to maintain constant technical actions, athletes needed to expend more energy in the last two rounds compared to the first, indicating a decrease in efficiency. Regarding relative energy contribution (Figure 2), the only significant change observed was a lower (*p* < 0.05) participation of the glycolytic system in the first round (7 ± 2%) compared with the last round (3 ± 3%). It is noteworthy that the ATP-PCr system had the same absolute and relative contributions across the three rounds, indicating that the 1 min intervals between rounds were sufficient for proper resynthesis of the phosphocreatine (PCr) stores. The high-intensity taekwondo actions are mainly supported by the ATP-PCr system; thus, its maintenance throughout the rounds likely kept the number of attacks, attack time, and sum of attack time constant during the entire match.

Figure 3 shows the relative energy system contributions during a simulated boxing match.

Davis et al. [14] estimated the energy system contributions during a simulated Olympic boxing match (Figure 3) and reported a high oxidative contribution of 86%, followed by ATP-PCr contribution of 10%, and glycolytic contribution of 4% (no standard deviations were provided). The total energy expenditure was 608.6 ± 81.8 kJ. The authors also reported that boxers achieved 97–100% of their peak oxygen consumption (VO_2peak_) during the last 20 s of each round, indicating the importance of well-developed aerobic power for these athletes. This is almost double the percentage achieved by karate athletes investigated by Doria et al. [15], indicating a marked difference between these striking combat sports in terms of oxygen uptake during the match. Moreover, Davis et al. [14] found that the absolute oxidative contribution was lower (*p* < 0.05) in the first round (126.8 ± 20.3 kJ) compared to the second (141.6 ± 24.0 kJ) and third (140.9 ± 28.6 kJ) rounds, whereas the absolute glycolytic contribution was higher (*p* < 0.05) in the first round (13.5 ± 4.1 kJ) compared to the second (8.4 ± 2.5 kJ) and third (4.3 ± 3.5 kJ) rounds. Values in the second round were also higher (*p* < 0.05) than those in the third round for this energy system. Vertical hip movements, total defense, and activity rate did not differ between rounds (Round 1: vertical hip movements = 86.6 ± 25.0 reps; total defense = 20.1 ± 7.3 reps; activity rate = 1.2 ± 0.2 actions/s; Round 2: vertical hip movements = 83.2 ± 10.7 reps; total defense = 19.2 ± 4.3 reps; activity rate = 1.2 ± 0.1 actions/s; Round 3: vertical hip movements = 90.0 ± 8.0 reps; total defense = 19.2 ± 5.0 reps; activity rate = 1.3 ± 0.1 actions/s), but total attacks were higher (*p* < 0.05) in the third round (47.0 ± 6.5 reps) compared to the second round (44.5 ± 6.8 reps), and total attacks executed during the first round (45.0 ± 3.7 reps) did not differ from the others. Despite some changes in the energy system contributions (e.g., decreased glycolytic contribution), boxers were able to increase the number of attacks in the third round compared to the second, but all other time-motion variables did not vary.

Therefore, these studies on striking combat sports have presented information on the total energy expenditure, metabolic power, and energy system contributions, and the relationship between metabolic demand and time-motion variables. This information is relevant to coaches, sport scientists, and strength and conditioning professionals, as specificity can be achieved more precisely when taking these factors into account. A few studies have also used this method to investigate the use of ergogenic aids [51,52] and to verify the impact of rapid weight loss [54], which is a common procedure among combat sport athletes [63], on the energy system contributions, total energy expenditure, and time-motion variables.

For instance, Lopes-Silva et al. [51,52] estimated the energy contribution of each metabolic pathway in taekwondo matches and aimed to enhance the lower contributing energy system (i.e., glycolytic) through the use of ergogenic substances (caffeine and sodium bicarbonate). It is noteworthy that Beneke et al. [12] found a correlation between higher glycolytic activation and increased net-action rate during karate matches. Based on these findings, Lopes-Silva et al. [51,52] hypothesized that the use of these substances would lead to a boost in glycolytic activation and thus, result in a greater number of high-intensity actions or extended periods of executing such actions. For the caffeine supplementation study [51], it was found that compared with a placebo, the use of caffeine increased (*p* < 0.05) the estimated glycolytic contribution (placebo = 8.9 ± 1.2 KJ; caffeine = 12.5 ± 1.7 kJ), but there were no changes in the number of attacks (placebo = 27.3 ± 2.1 attacks; caffeine = 26.7 ± 1.9 attacks) or attack time (placebo = 36.6 ± 4.5 s; caffeine = 33.8 ± 1.9 s). Therefore, the increased glycolytic contribution did not result in higher taekwondo performance.

However, sodium bicarbonate supplementation was more effective in enhancing performance during the simulated taekwondo match [52]. The supplementation resulted in a higher (*p* < 0.05) glycolytic activation (31% increase) in the first round (12.0 ± 3.9 KJ) compared to the placebo (8.6 ± 3.2 KJ), and a longer (13% increase; *p* < 0.05) sum of attack time (25.7 ± 9.0s) compared with the placebo (22.0 ± 7.3s). Therefore, the increase in the sum of attack time was likely due to the higher glycolytic activation, suggesting that sodium bicarbonate ingestion could be a useful ergogenic aid for taekwondo athletes.

Combat sports athletes often reduce their body mass to compete against potentially smaller and weaker opponents. This process is called rapid weight loss and is typically followed during the week leading up to the competition using various methods to induce dehydration [63]. In taekwondo, 75.6% and 88.6% of male and female athletes at the national and international level, respectively, reported using rapid weight loss procedures to compete in a lighter weight category [64]. These athletes typically lose 3% of their body mass, with some reporting a 7% reduction in the week prior to competition [64]. While there is evidence that rapid weight loss can harm an athlete’s health and performance [63], a recent meta-analysis suggested that combat sport athletes can handle reductions of up to 5% of their body mass when they have over 3 h to recover after weigh-in [65]. Currently, taekwondo athletes weigh-in the night before competition; therefore, recovery time is around 15–17 h [66].

Yang et al. [54] evaluated the energy system contributions during a simulated taekwondo competition that consisted of three matches separated by one hour, both in control and rapid weight loss conditions. In the rapid weight loss condition, athletes lost 5% of their body mass over three and a half days and had 16 h to recover between weigh-in and the simulated competition. In the control condition, no rapid weight loss was conducted. The only significant difference found was a higher absolute and relative glycolytic contribution in the third match of the rapid weight loss condition compared to the control condition (*p* < 0.05) (glycolytic contribution in rapid weight loss condition = 8.6 ± 4.2 kJ, 8 ± 3%; in control condition = 5.6 ± 4.7 kJ, 5 ± 4%). However, as the athletes performed more kick attacks during bouts 1 and 3 in the rapid weight loss condition (*p* < 0.05) (bout 1 in rapid weight loss condition = 20.73 ± 5.10 kicks; bout 3 = 21.73 ± 6.31 kicks; bout 1 in control condition = 15.73 ± 5.83 kicks; bout 3 = 16.53 ± 3.44 kicks), it is likely that the increase in glycolytic activation was due to a higher number of attacks. With 16 h to recover after weigh-in, this study showed that rapid weight loss does not harm taekwondo athletes’ performance and that more offensive actions can result in higher glycolytic activation compared to the control condition. Therefore, rapid weight loss appears to benefit taekwondo athletes’ performance.

### 2.2. Grappling Olympic Combat Sports

Figure 4 presents the relative energy system contributions during simulated judo matches lasting from 1 min up to 5 min.

The only study estimating energy system contributions in grappling combat sport was performed with judo athletes [16]. They found that the oxidative system dominated (79%), followed by the ATP-PCr (14%) and glycolytic (7%) systems during a 4 min match (Figure 4), which is the official match duration. Energy expenditure during this duration was 350 ± 91 kJ. In this study [16], matches lasting 1, 2, 3, 4 and 5 min were analyzed, as in judo matches can finish before the time limit when one athlete scores an ippon or may have an extra-time when the 4 min results in a tie, in which case the athlete who scores first wins [67]. Julio et al. [16] also determined the onset of blood lactate accumulation (OBLA) and VO_2peak_ for upper- and lower-body cycle ergometer. Oxygen consumption was higher during the match than at upper-body OBLA for all durations, and higher than lower-body OBLA for 3- and 4 min matches, but slightly lower for 1, 2 and 5 min matches. When upper-body cycle ergometer VO_2peak_ was considered, match values were close only for the 4 min duration, whereas values of~80% of VO_2peak_ were achieved for all other durations. The match values were 60% to 75% of lower-body VO_2peak_. Thus, during a match, judo athletes seem to consume oxygen around their upper-body OBLA but lower than their VO_2peak_. The authors reported lower absolute oxidative contribution in 1 min matches compared to 2 to 5 min (*p* < 0.05), lower values in 2 min matches compared to longer durations (*p* < 0.001), and lower (*p* < 0.001) values in 3 min matches compared with longer durations. The relative oxidative contribution was lower in 1 min matches compared to other durations (*p* < 0.001), and lower in 2 min matches compared to longer matches (*p* < 0.001). The relative ATP-PCr contribution was higher in 1 min matches compared with other durations (*p* < 0.001) and higher in 2 min matches compared to longer matches (*p* < 0.05). Longer matches are more oxidative and rely less on the ATP-PCr system, whereas the glycolytic system remains unchanged in judo matches lasting 1 to 5 min. Consequently, absolute metabolic power also varied in matches of different durations, with higher values in 1 min matches compared to longer matches (*p* < 0.001). Therefore, interventions that minimize the decrease in metabolic power during a match could improve performance. This is especially relevant when considering the findings reported by the authors regarding lower (*p* < 0.05) pause time per sequence adopted by athletes in 1 min match compared with the 5 min match, suggesting less need for recovery in shorter matches due to higher metabolic power at the beginning of the match.

### 2.3. Weapon-Based Combat Sport

Figure 5 presents the relative energy system contributions during three 3 min fencing épée matches, and the mean contributions throughout the entire match.

In weapon-based combat sports, Yang et al. [18] recently estimated the energy system contributions during a simulated fencing épée match consisting of three 3 min bouts separated by 1 min intervals. They did not present overall energy system contributions or metabolic power for the entire match, instead, they highlighted the main findings for each bout. Energy expenditure remained consistent between successive bouts (bout 1: 230.67 ± 41.84 kJ; bout 2: 206.19 ± 30.17 kJ; bout 3: 210.21 ± 28.96 kJ). Oxidative metabolism provided the highest absolute and relative contribution (*p* < 0.001) to total energy expenditure, followed by the ATP-PCr and glycolytic systems. Specifically, when considering the absolute contribution, the oxidative system (bout 1: 185.55 ± 36.43 kJ; bout 2: 185.46 ± 28.81 kJ; bout 3: 186.62 ± 27.70 kJ) and ATP-PCr system (bout 1: 28.70 ± 10.09 kJ; bout 2: 19.53 ± 9.49 kJ; bout 3: 22.25 ± 9.27 kJ) did not fluctuate between bouts, whereas the glycolytic system showed a higher contribution (*p* < 0.01) in the first bout (16.42 ± 6.47 kJ) compared to bouts 2 (1.20 ± 1.32 kJ) and 3 (1.34 ± 1.84 kJ). As for the relative energy contribution, the oxidative system had a lower contribution (*p* < 0.05) in the first bout (80.57 ± 4.45%) compared to the second (90.02 ± 4.69%) and third bouts (88.8 ± 4.28%), whereas the glycolytic system had a higher contribution (*p* < 0.01) in the first bout (6.97 ± 2.53%) compared to the second (0.63 ± 0.73%) and third bouts (0.60 ± 0.80%). The ATP-PCr system remained constant between bouts (bout 1: 12.44 ± 6.67%; bout 2: 9.35 ± 4.30%; bout 3: 10.60 ± 4.52%). However, despite these variations in energy system contributions, no changes were observed in the number of attack and defensive actions or the time spent on them in the three bouts. Thus, since the energy expenditure remained constant across the three bouts, as noted above, the lack of difference in these technical and time-motion variables suggests that different combinations of energy system contributions were effective in preserving performance throughout the bouts. The authors also found significant positive correlations between oxidative contributions and the combined attack and defense times (r = 0.49), indicating that athletes relying more on this system were able to spend more time attacking and defending. Given the total duration of the simulated match (9 min) and the fact that the PCr stores are replenished through oxidative processes, this pathway is crucial for maintaining high-intensity actions during fencing matches.

## 3. Conclusions

Performance in combat sports is influenced by both physiological and technical abilities; athletes with higher energy output are likely to perform more intense actions or repeat them more frequently, but at the same time, athletes who are more efficient can perform the same action with less energy expenditure. Understanding the interaction between energy system contributions, total energy expenditure, and metabolic power with technical actions performed by combat sports athletes during match simulations provides valuable information for adjusting training programs. For example, the estimate of energy system contributions can be used to track an athlete’s physiological adaptation to different training methods and interventions such as ergogenic aids, warm-up techniques, rapid weight loss procedures, and recovery strategies between matches. However, information regarding the athletes’ physical fitness and its influence on the energy system contributions in response to their actions is still incipient. Therefore, comparing athletes from different weight categories and with unique combat styles could contribute to a better understanding of the metabolic responses in these specific situations. Wrestling has not been investigated yet, and there is a need for studies to estimate the energy systems’ contribution in this sport. Additionally, only one study [16] has investigated partial time during a typical combat. This approach is likely to have been used in judo as it is more common for matches to be finished before the full time in this sport compared to other combat sports. However, for other combat sports involving rounds or bouts longer than three minutes, the information on the energy systems’ contributions during shorter periods could provide valuable insights. This paper addressed energy system contributions in different Olympic combat sports and serves as a starting point for further research to improve training methods and performance in these sports.

## Figures and Tables

**Figure 1 metabolites-13-00297-f001:**
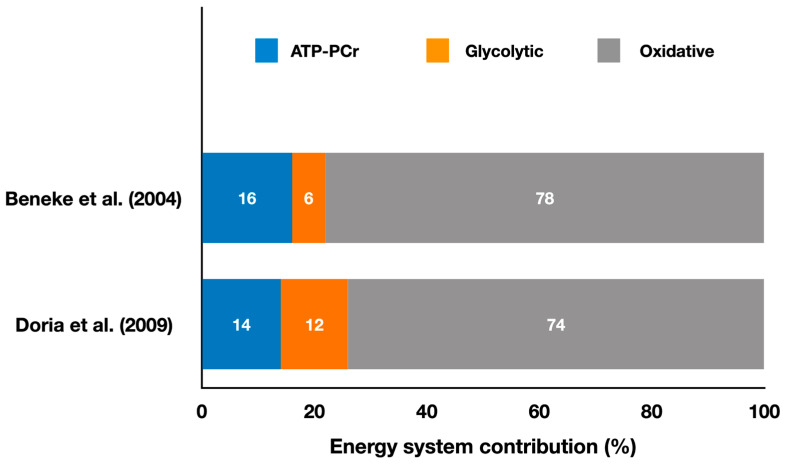
Contributions of the oxidative, ATP-PCr, and glycolytic systems during simulated karate matches (adapted from Beneke et al. [12] and Doria et al. [15]). The values represent the mean percentage of each energy system throughout the entire match simulation (refer to the text for specific comparisons between energy systems in each study).

**Figure 2 metabolites-13-00297-f002:**
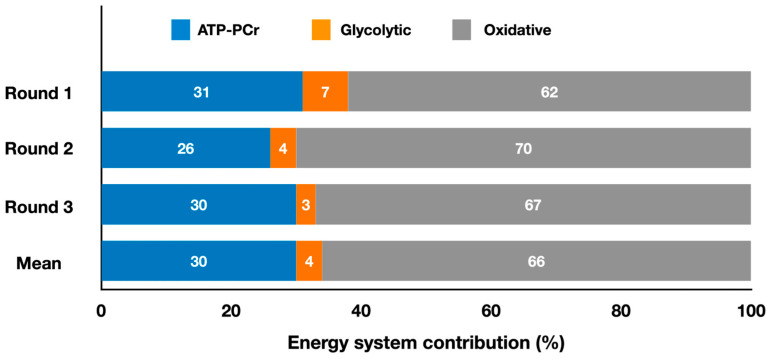
Contributions of the oxidative, ATP-PCr, and glycolytic systems during three 2 min rounds of a taekwondo simulated match, and the mean contributions throughout the entire match (adapted from Campos et al. [13]). The values represent the mean percentage of each energy system for each round and throughout the entire match simulation (refer to the text for specific comparisons between energy systems).

**Figure 3 metabolites-13-00297-f003:**
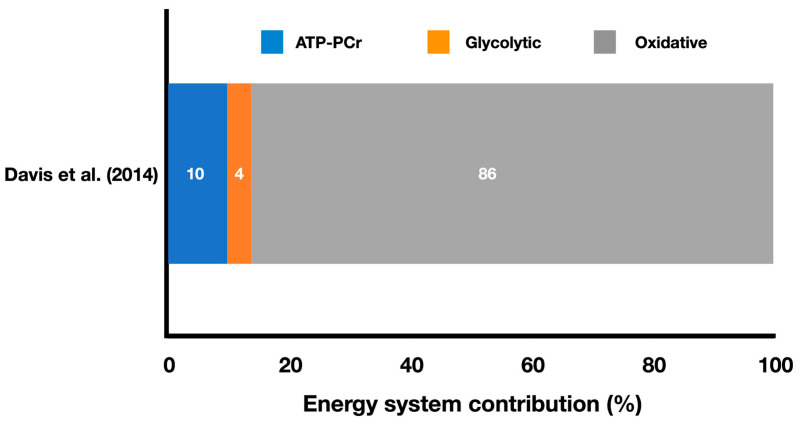
Contributions of the oxidative, ATP-PCr, and glycolytic systems during a boxing simulated match (adapted from Davis et al. [14]). The values represent the mean percentage of each energy system throughout the entire match simulation (refer to the text for specific comparisons between energy systems).

**Figure 4 metabolites-13-00297-f004:**
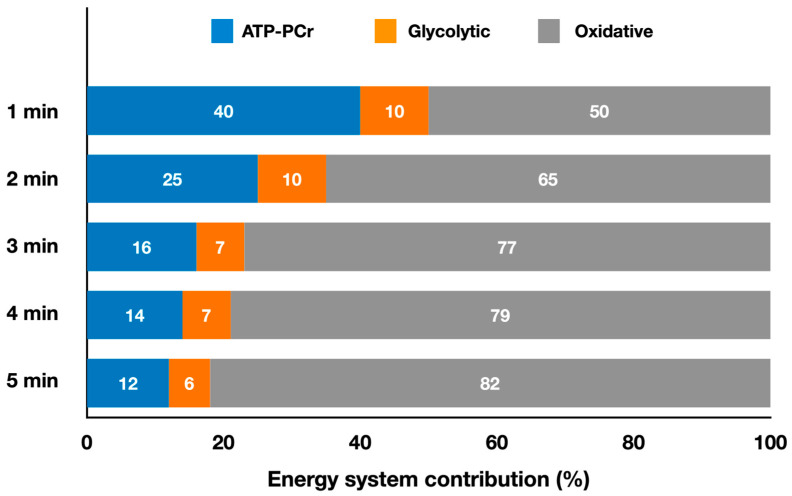
Contributions of the oxidative, ATP-PCr, and glycolytic systems during judo simulated matches lasting from 1 min up to 5 min (adapted from Julio et al. [16]). The values represent the mean percentage of each energy system for each duration (refer to the text for specific comparisons between different match durations).

**Figure 5 metabolites-13-00297-f005:**
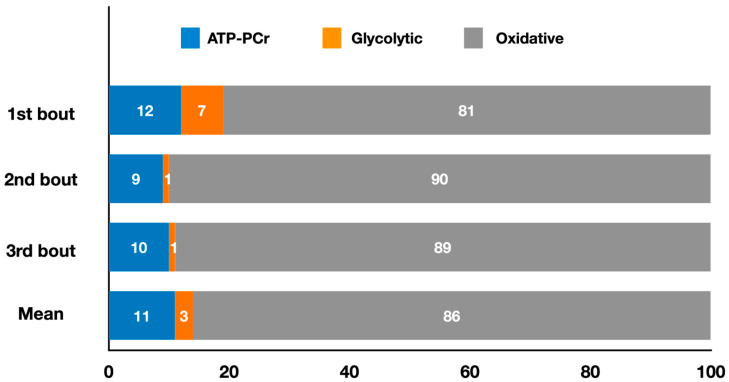
Contributions of the oxidative, ATP-PCr, and glycolytic systems during three 3 min fencing épée matches, and the mean contributions throughout the entire match (adapted from Yang et al. [18]). The values represent the mean percentage of each energy system for each round and for the entire match simulation (refer to the text for specific comparisons between energy systems).

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
