# Peer review of "Energy System Contributions during Olympic Combat Sports: A Narrative Review"

_metabolites, 2023, doi:10.3390/metabo13020297_

Round 1

Reviewer 1 Report

The description in this manuscript is poorly explained and the message is unclear. Therefore, it is not possible to accept this manuscript.  Recommendation; Rewrite and resubmittion.

Comments

1. Although the three energy systems are the main content of this paper, there is no explanation for the three systems.

2. Please explain the following three contents using the figure. 1) oxidative system, 2) the ATP-PCr system, 3) glycolytic systems. The authors must explain how the three energy systems perform their analysis and what each reflects.

3. The figure 1 is not informative. Please rewrite the figure 1..

4. I don't understand why this review is necessary as the conclusion is so unfocused.

Author Response

Dear Guest Editors

I would like to thank you and the reviewers for giving me the opportunity to improve the content of this paper. I have presented my responses to each editor's or reviewer's comments below. I hope that I have properly addressed the points raised and that the article has reached a good level for publication in this journal. I also appreciate any further comments you may have.

Best regards,

Emerson Franchini 

Reviewer 1

The description in this manuscript is poorly explained and the message is unclear. Therefore, it is not possible to accept this manuscript.  Recommendation; Rewrite and resubmittion.

Comments

1. Although the three energy systems are the main content of this paper, there is no explanation for the three systems.

Reply: I would like to express my gratitude to the reviewer for her/his insightful comment. In order to address this issue, I have included a brief explanation of each energy system in the introduction section. This part now reads as: "Briefly, three energy systems contribute to adenosine triphosphate (ATP) resynthesis and thus to energy release during exercise, as only the hydrolysis of the ATP molecule provides energy for skeletal muscle contraction. These systems are the ATP-PCr (adenosine triphosphate and phosphocreatine), glycolytic, and oxidative systems [46]. The ATP-PCr and glycolytic systems are classified as anaerobic, while the oxidative system is classified as aerobic. The ATP-PCr system has a higher metabolic power (rate of energy transfer per unit of time) due to a low number of reactions needed to resynthesize ATP, but a low capacity (total amount of energy that can be released) due to limited substrate stores. The glycolytic system resynthesizes ATP through the non-aerobic breakdown of carbohydrates, mainly stored as muscle glycogen, and has intermediate metabolic power due to the higher number of reactions involved compared to the ATP-PCr system, but a greater capacity due to a large amount of stored carbohydrates. However, it is mainly limited by the accumulation of metabolites during activation. The oxidative system depends on oxygen consumption and degrades carbohydrates, fats, and sometimes protein. It has the lowest metabolic power among the energy systems due to many reactions involved in the oxidative breakdown of these substrates, but it has the highest capacity as the stores of these substrates are abundant. The activation of these energy systems determines the rate of energy release and thus the intensity and duration of effort [46].”

2. Please explain the following three contents using the figure. 1) oxidative system, 2) the ATP-PCr system, 3) glycolytic systems. The authors must explain how the three energy systems perform their analysis and what each reflects.

Reply: An explanation of each energy system was included in the text (please, see the comment above). I kindly request that this remain in text form only, as new figures have already been incorporated into the text.

3. The figure 1 is not informative. Please rewrite the figure 1..

Reply: Given the nature of this review, Figure 1 was altered and additional figures were inserted. Each figure presents information reporting the percentage of energy system contributions and is accompanied by a written explanation. I hope these changes have addressed this issue. I would be grateful for any specific suggestions regarding this matter. 

4. I don't understand why this review is necessary as the conclusion is so unfocused.

Reply: The rationale for this review is primarily outlined in the introduction section. Additionally, new elements were added in the conclusion section.

Reviewer 2 Report

This is a timely and interesting review that has a very novel and relatively understudied perspective.  Moreover, there is a laudable coverage of the extant literature.  That said, there are several aspects that might add greater understanding, applicability and relevance to understanding the energetics principles considered.  Paramount among these are:

1.       Kinetics.  The participation of oxidative processes to meet the energetics demands of a given metabolic transition will be dependent upon the speed of the VO2 kinetics as well as the number and size of metabolic transients considered.  A brief tutorial on kinetics would be appropriate herein.  Also, as VO2 kinetics relates directly to overall fitness – VO2max (see Rossiter, 2011; Poole & Jones, 2012) – such data on these athletes would be relevant.

2.       As Professor Franchini is doubtless aware, within each sport/art the different weight categories and styles add a substantial heterogeneity of energetics that is missed in the mean data.  In Shotokan karate, for example, the very static counterpuncher Billy Higgins – in his day – versus the mobile Andy Sherry or Richard Poole in the UK, for instance. In judo, the grappler and groundwork specialist Anton Geesinck versus the throwing specialists – Japanese – or superb combination fighters such as Neil Adams and Brian Jacks, in their day. A brief consideration of such heterogeneity would be greatly appreciated.

3.       Perhaps too much weight is given to blood [lactate] measurements given the assumptions regarding body water distribution in the estimation of “anaerobic contribution.”  I believe the VO2 equivalent is from the old estimations of Margaria and we know a lot more about the inter-compartmental movements of lactate nowadays thanks to the excellent work of George Brooks and many others.

4.       For clarity please label the sports on each panel so the reader can appreciate the data at a glance.

Rossiter HB. Exercise: Kinetic considerations for gas exchange. Compr Physiol. 2011 Jan;1(1):203-44. doi: 10.1002/cphy.c090010. PMID: 23737170.

Poole DC, Jones AM. Oxygen uptake kinetics. Compr Physiol. 2012 Apr;2(2):933-96. doi: 10.1002/cphy.c100072. PMID: 23798293.

Author Response

Reviewer 2

This is a timely and interesting review that has a very novel and relatively understudied perspective.  Moreover, there is a laudable coverage of the extant literature.  That said, there are several aspects that might add greater understanding, applicability and relevance to understanding the energetics principles considered.  Paramount among these are:

1.       Kinetics.  The participation of oxidative processes to meet the energetics demands of a given metabolic transition will be dependent upon the speed of the VO2 kinetics as well as the number and size of metabolic transients considered.  A brief tutorial on kinetics would be appropriate herein.  Also, as VO2 kinetics relates directly to overall fitness – VO2max (see Rossiter, 2011; Poole & Jones, 2012) – such data on these athletes would be relevant.

Reply: Thank you for this suggestion. I have taken into consideration the kinetics by incorporating the article from Rossiter (2011). This added section reads as follows: "Moreover, this method does not require quantifying the mechanical output, and it allows estimating the energy systems by measuring continuous oxygen uptake during simulations of combat sport matches [9]. This is particularly relevant because during intense, intermittent efforts, such as those performed in combat sports, the gas exchange is entirely kinetic, meaning that the dynamics of pulmonary oxygen uptake and pulmonary carbon dioxide production never reach a steady state. This because their kinetics are slow relative to the imposed effort [55]."

2.       As Professor Franchini is doubtless aware, within each sport/art the different weight categories and styles add a substantial heterogeneity of energetics that is missed in the mean data.  In Shotokan karate, for example, the very static counterpuncher Billy Higgins – in his day – versus the mobile Andy Sherry or Richard Poole in the UK, for instance. In judo, the grappler and groundwork specialist Anton Geesinck versus the throwing specialists – Japanese – or superb combination fighters such as Neil Adams and Brian Jacks, in their day. A brief consideration of such heterogeneity would be greatly appreciated.

Reply: Thank you for this comment. A brief sentence regarding this aspect was inserted in the conclusion section: However, information regarding the athletes’ physical fitness and its influence on the energy system contributions in response to their actions is still incipient. Therefore, comparing athletes from different weight categories and with unique combat styles could contribute to a better understanding of the metabolic responses in these specific situations.

3.       Perhaps too much weight is given to blood [lactate] measurements given the assumptions regarding body water distribution in the estimation of “anaerobic contribution.”  I believe the VO2 equivalent is from the old estimations of Margaria and we know a lot more about the inter-compartmental movements of lactate nowadays thanks to the excellent work of George Brooks and many others.

Reply: Thank you for this comment. I agree that there may be some limitations to using this approach. As mentioned in the "Energy System Contributions in Combat Sports" topic, this was the only method used for this purpose in combat sports ("Each method has its own assumptions and limitations, which are beyond the goals of the present review, but some cannot be used to estimate the energy system contributions when the efficiency or the mechanical power/speed cannot be quantified, whereas others can be used during combat sports match simulations. Indeed, the method involving the measurement of oxygen uptake, EPOCfast and blood lactate has been the solely method used to estimate the energy system contributions in combat sports”). In light of potential criticism regarding the validity of this method, a specific paragraph has been added to the topic that cites studies published in the last 15 years, with a focus on articles published in the last 5 years. This paragraph reads as follows: "Moreover, this method does not require quantifying the mechanical output, and it allows estimating the energy systems by measuring continuous oxygen uptake during simulations of combat sport matches [9]. This is particularly relevant because during intense, intermittent efforts, such as those performed in combat sports, the gas exchange is entirely kinetic, meaning that the dynamics of pulmonary oxygen uptake and pulmonary carbon dioxide production never reach a steady state. This because their kinetics are slow relative to the imposed effort [55]."

4.       For clarity please label the sports on each panel so the reader can appreciate the data at a glance.

Reply: Thank you for your suggestion. Based on your comment and others received, new figures were incorporated to improve the article. If you have any further specific suggestions, they would be greatly appreciated.

Reviewer 3 Report

Dear Author

You have written an interesting narrative review paper discussing Energy system contributions during Olympic combat sports.

At the end of the introduction section, I am missing the inclusion of wrestling in the inclusion criteria. Is there any specific reason this Olympic combat sport was not included? Please elaborate and amend.

Overall a straightforward review paper with on-point and current research with good English and structure.

Nonetheless, in my opinion, wrestling should be included in this review. Therefore, I recommend a major revision.

Kind regards

Author Response

Reviewer 4

You have written an interesting narrative review paper discussing Energy system contributions during Olympic combat sports. 

Reply: Thank you for this comment.

At the end of the introduction section, I am missing the inclusion of wrestling in the inclusion criteria. Is there any specific reason this Olympic combat sport was not included? Please elaborate and amend.

Reply: Thank you for this relevant comment. The term was inserted, and an explanation regarding the absence of studies estimating the energy system contributions during simulated wrestling matches through physiological measurements was added to the end of the introduction section.

Overall a straightforward review paper with on-point and current research with good English and structure.

Reply: thank you for this comment.

Nonetheless, in my opinion, wrestling should be included in this review. Therefore, I recommend a major revision.

Reply: Thank you for the opportunity to improve this article. The explanation about the lack of wrestling studies was added. 

Round 2

Reviewer 1 Report

The author refined the manuscript, however, consideration is superficial. 

Please refine the manuscript.

1. Please describe the energy system contributions during relaxed state before the game. Do different sports have different energy systems in a relaxed state before a match?

2. There is an energy system that matches the physique of each athletes. For example, in judo, is the energy system the same for lightweight and heavyweights?

3. Judo shows a dynamic energy system change during the match. Please discuss about what advantages and challenges it has compared to other sports.

4. Please add description about the energy system contributions during resting and exercise in non-athlete.

5. All five figures are not informative. Please rewrite these figures. I have attached five figures that I created based on the data in this manuscript. Please create the same format as these figures. 

Author Response

Dear Guest Editors

I would like to thank you and the reviewers for giving me the opportunity to improve the content of this paper. I have presented my responses to the reviewer's comments below. I hope that I have properly addressed the points raised and that the article has reached a good level for publication in this journal. I also appreciate any further comments you may have.

Best regards,

Emerson Franchini 

Reviewer 1

The author refined the manuscript, however, consideration is superficial. 

Please refine the manuscript.

1. Please describe the energy system contributions during relaxed state before the game. Do different sports have different energy systems in a relaxed state before a match?

Reply: The activation of energy systems is dependent on the intensity and duration of the effort performed. During rest, the metabolic demand is primarily met through the oxidative system. Athletes perform warm-ups before competing to prepare for competition, but the impact of different warm-ups on the energy systems' contributions has not been investigated, especially for athletes from different sports. The relevance of studies addressing this aspect is suggested in the conclusion section: "For example, the estimate of energy system contributions can be used to track an athlete's physiological adaptation to different training methods and interventions such as ergogenic aids, warm-up techniques, rapid weight loss procedures, and recovery strategies between matches”.  

2. There is an energy system that matches the physique of each athletes. For example, in judo, is the energy system the same for lightweight and heavyweights?

Reply: It is likely that small variations may exist among judo athletes from different weight categories. The actions executed during a match, such as more or less aggressive styles, are likely to have a greater impact on energy systems activation than the athletes' body mass. However, this has not yet been investigated. This was also suggested in the conclusion section: “… information regarding the athletes’ physical fitness and its influence on the energy system contributions in response to their actions is still incipient. Therefore, comparing athletes from different weight categories and with unique combat styles could contribute to a better understanding of the metabolic responses in these specific situations.” 

3. Judo shows a dynamic energy system change during the match. Please discuss about what advantages and challenges it has compared to other sports.

Reply: In fact, the study by Julio et al. (2017) was the only one to investigate the different durations of matches and their effect on energy system contributions. The significance of this approach was emphasized in the conclusion section, which now reads as: "Additionally, only one study [16] has investigated partial time during a typical combat. This approach is likely to have been used in judo as it is more common for matches to be finished before the full time in this sport compared to other combat sports. However, for other combat sports involving rounds or bouts longer than three minutes, the information on the energy systems’ contributions during shorter periods could provide valuable insights."

4. Please add description about the energy system contributions during resting and exercise in non-athlete.

Reply: As indicated in response to suggestion 1, the energy systems are activated based on the intensity and duration of the effort. As a result, it is not feasible to provide a general response to exercise in non-athletes, which is not relevant to the current review's topic of exploring the energy system contributions in various combat sports. Therefore, I kindly request to keep the text as presented.

5. All five figures are not informative. Please rewrite these figures. I have attached five figures that I created based on the data in this manuscript. Please create the same format as these figures. 

Reply: Graphs were changed as requested.

Reviewer 3 Report

Dear Author,

Thank you for addressing my comments. The overall quality of the manuscript improved.

Therefore, I recommend acceptance in its current form.

Kind regards

Author Response

Dear Guest Editors

I would like to thank you and the reviewers for giving me the opportunity to improve the content of this paper. I have presented my responses to the reviewer's comments below. I hope that I have properly addressed the points raised and that the article has reached a good level for publication in this journal. I also appreciate any further comments you may have.

Best regards,

Emerson Franchini 

Reviewer 3

Dear Author,

Thank you for addressing my comments. The overall quality of the manuscript improved.

Therefore, I recommend acceptance in its current form.

Reply: Thank you for your suggestions. 
